# Evaluation and Application of Drug Resistance by Biomarkers in the Clinical Treatment of Liver Cancer

**DOI:** 10.3390/cells12060869

**Published:** 2023-03-10

**Authors:** Po-Shuan Huang, Ling-Yu Wang, Yi-Wen Wang, Ming-Ming Tsai, Tzu-Kang Lin, Chia-Jung Liao, Chau-Ting Yeh, Kwang-Huei Lin

**Affiliations:** 1Graduate Institute of Biomedical Sciences, College of Medicine, Chang Gung University, Taoyuan 333, Taiwan; leo_6813@msn.com (P.-S.H.); l329735@ms49.hinet.net (C.-J.L.); 2Department of Biochemistry and Molecular Biology, Chang Gung University, Taoyuan 333, Taiwan; lywang@mail.cgu.edu.tw; 3Division of Hematology-Oncology, Chang Gung Memorial Hospital at Linkou, Taoyuan 333, Taiwan; 4School of Nursing, College of Medicine, Chang Gung University, Taoyuan 333, Taiwan; wangyw@mail.cgu.edu.tw; 5Department of Nursing, Division of Basic Medical Sciences, Chang Gung University of Science and Technology, Taoyuan 333, Taiwan; mmtsai@mail.cgust.edu.tw; 6Research Center for Chinese Herbal Medicine, College of Human Ecology, Chang Gung University of Science and Technology, Taoyuan 333, Taiwan; 7Department of General Surgery, New Taipei Municipal Tu Cheng Hospital, New Taipei 236, Taiwan; 8Neurosurgery, School of Medicine, College of Medicine, Fu Jen Catholic University, New Taipei City 24205, Taiwan; tklin100@cgmh.org.tw; 9Neurosurgery, Department of Surgery, Fu Jen Catholic University Hospital, New Taipei City 24352, Taiwan; 10Liver Research Center, Chang Gung Memorial Hospital, Linkou, Taoyuan 333, Taiwan; chautingy@gmail.com

**Keywords:** liver cancer, biomarker, clinical, drug resistance

## Abstract

Liver cancer is one of the most lethal cancers in the world, mainly owing to the lack of effective means for early monitoring and treatment. Accordingly, there is considerable research interest in various clinically applicable methods for addressing these unmet needs. At present, the most commonly used biomarker for the early diagnosis of liver cancer is alpha-fetoprotein (AFP), but AFP is sensitive to interference from other factors and cannot really be used as the basis for determining liver cancer. Treatment options in addition to liver surgery (resection, transplantation) include radiation therapy, chemotherapy, and targeted therapy. However, even more expensive targeted drug therapies have a limited impact on the clinical outcome of liver cancer. One of the big reasons is the rapid emergence of drug resistance. Therefore, in addition to finding effective biomarkers for early diagnosis, an important focus of current discussions is on how to effectively adjust and select drug strategies and guidelines for the treatment of liver cancer patients. In this review, we bring this thought process to the drug resistance problem faced by different treatment strategies, approaching it from the perspective of gene expression and molecular biology and the possibility of finding effective solutions.

## 1. Introduction

Primary liver cancer is the second leading cause of cancer-related death worldwide, and as such is a major public health challenge [1]. Historically, the main factors implicated in causing liver cancer are excess alcohol consumption and viruses. In the latter context, chronic infection with hepatitis B virus (HBV) and hepatitis C virus (HCV) are the greatest etiological risk factors for hepatocellular carcinoma (HCC) [2]. As medical principles and practices have evolved, the risk factors for liver cancer have come into greater focus, and it has even begun to be suggested that liver cancer should be considered a metabolic disease. At the same time, liver cancer has gradually become more serious as a result of changes in people’s daily life, with obesity, type 2 diabetes, and nonalcoholic fatty liver disease (NAFLD) replacing viral- and alcohol-related liver disease as major pathogenic promoters [3]. The fact that there are natural genetic determinants of liver cancer cannot be ignored. As is the case for most other cancers, the expression of the patient’s own genes reflects—and is reflected in—many factors, including the course of liver cancer, the degree of associated morbidity, and the survival rate [4]. Against this backdrop, it is logical to discuss treatment strategies in terms of the patient’s genotype. After all, treatments such as targeted therapy and immunotherapy are highly genotype-related interventions [5,6]. In the context of immunotherapy, whether a patient’s liver cancer cells highly express cytotoxic T lymphocyte-associated antigen 4 (CTLA-4), programmed cell death protein-1 (PD1), programmed cell death ligand 1 (PD-L1), or other immune checkpoint genes that prevent immune cells from functioning is key to whether treatment with antibodies is likely to be effective [7]. In addition, antibody-based biologics such as sorafenib and Lenvatinib, which act on multiple receptor tyrosine kinases (RTKs), are also based on the expression of abnormal genes in cancer cells [8,9]. Sorafenib can significantly extend the median survival time of patients, but only by 5 months. It is also associated with serious adverse side effects as well as the frequent development of drug resistance [10]. In light of these diverse drug-resistance phenomena, it is critical to understand the various underlying cellular mechanisms involved and employ suitable biomarkers for making drug-matching or -replacement decisions.

## 2. Clinical Treatment Strategies for Liver Cancer

There are many different methods for the staging of liver cancer in clinical practice, including Okuda, Barcelona Clinic Liver Cancer (BCLC), Cancer of the Liver Italian Program (CLIP), Chinese University Prognostic Index (CUPI), Japan Integrated Staging (JIS), and tumor–node–metastasis (TNM), among others. Although the details differ among methods, most are based on the size of the tumor and its metastasis status [11,12]. Clinically, these different staging methods are highly correlated with treatment strategies for liver cancer patients (Figure 1). According to the 2022 BCLC treatment strategy, patient evaluation should take into account both tumor status and performance status (PS) as well as liver function (Child–Pugh score). For early-stage patients (0-A), liver resection or ablation (radiofrequency or microwave) may also be accepted after evaluation for liver transplantation. In intermediate-stage patients (B), besides liver transplantation, patients will be evaluated for transarterial chemoembolization (TACE) and systemic drug therapy, including targeted drugs and immunotherapy. Advanced-stage patients (C), should receive systemic drug treatment first, but other options may be considered after evaluation. As for patients with end-stage liver cancer (D), they can only receive supportive therapy, as various HCC therapies are no longer effective [13].

Sorafenib was the first targeted liver cancer drug to pass Phase III clinical trials and receive approval by the United States Food and Drug Administration (US-FDA) for the treatment of unresectable HCC and advanced renal cell carcinoma. Sorafenib is an oral multiple tyrosine kinase inhibitor (TKI) that inhibits angiogenesis, cell proliferation, and cell differentiation by targeting both the RAF/MEK/ERK pathway and receptor tyrosine kinases (Figure 2) [14]. Among the specific targets of this small-molecule inhibitor are the vascular endothelial growth factor receptor (VEGFR), platelet-derived growth factor (PDGFR), stem cell factor receptor (c-KIT), and Fms-like tyrosine-kinase receptor (FLT-3), among others [15,16]. By interfering with these signaling pathways, sorafenib has the potential to interrupt cellular processes in cancer cells, leading to apoptosis, cell death, and decreased cell proliferation [17]. Clinical trials have demonstrated that treatment of liver cancer with sorafenib can significantly improve the survival rate of patients with advanced liver cancer by about 5 months [18]. Unfortunately, despite this increased survival time, sorafenib does not significantly aid in controlling the cancer, likely owing to the development of drug resistance during the treatment process [19].

Lenvatinib is another targeted drug approved as first-line treatment, because its effects are, at minimum, comparable to those of sorafenib. Preclinical studies have shown that lenvatinib inhibits tumor proliferation and possesses immunomodulatory activity [20], with the most common treatment-emergent adverse events being hypertension, diarrhea, decreased appetite, and weight loss. Given that its efficacy is not inferior to that of sorafenib and its tolerability profile is manageable, lenvatinib represents a long-awaited alternative option to sorafenib for the first-line systemic treatment of patients with unresectable HCC [8].

In addition to drugs for first-line treatment, some second-line drugs for HCC treatment, namely regorafenib and cabozantinib, have received successive approvals in recent years. Although these compounds target different gene products, they are both TKI-type drugs, so their mechanism of action is still the same as that of sorafenib and lenvatinib [21]. In recent years, immunotherapeutic research has undergone a rapid transformation, giving rise to the development of the immunotherapy-based second-line HCC treatments nivolumab and pembrolizumab [22]. Among the many types of immunotherapy drugs, anti-PD1 agents such as nivolumab and pembrolizumab are the most important in the clinical treatment of HCC [23].

The human immune system is very powerful, with the ability to eliminate many invading pathogens and harmful cells. However, it has also developed a set of defense mechanisms to reduce damage caused by the immune mechanism to some normal cells through a pathway called immune checkpoints. Of all of the negative regulatory receptors that mediate these inhibitory feedbacks, PD1 has been one of the most intensively investigated owing to its indispensable role in fine-tuning the function of T cells and maintaining immune system homeostasis. Acting as a natural brake, PD1 is capable of eliciting an immune checkpoint response in T cells that is commonly associated with peripheral tolerance. However, tumor cells take advantage of this negative checkpoint regulation to suppress immunity and evade immune surveillance. Anti-PD1 agents, developed based on the proposed concept of “checkpoint blockade”, have been tested and shown to release the brakes on the immune system and unleash antitumor immune responses in cancer management [23].

Nivolumab is a type of anti-PD1 drug approved by the US FDA for advanced HCC following molecular targeted therapy with sorafenib. Nivolumab prevents the interaction of PD1 with the programmed death ligands, PD-L1 and PD-L2, blocking binding that would otherwise lead to a poor T-cell response toward HCC tumors [24]. Clinical studies have found that although nivolumab can prolong the survival of patients with advanced liver cancer, this aspect of its efficacy is not significantly better than that of sorafenib. However, nivolumab does show significantly reduced side effects (fatigue, abdominal pain, rash, cough, and decreased appetite) and accompanying tolerability issues compared with sorafenib [25]. Pembrolizumab, like nivolumab, is also a targeted immunotherapeutic drug that acts through basically the same principle as an anti-PD1 agent [26]. Approvals for clinical applications of drugs vary slightly, but differences mainly reflect inherent differences in their mechanisms of action or pharmacokinetic properties or discrepancies in clinical trial design [27]. However, as a practical matter, not all patients have the option of using immunotherapy drugs as their treatment strategy of choice. For example, how the tumor environment is affected by immune cells or whether the tumor type expresses PD1/PD-L1 will fundamentally affect the efficacy of immunotherapy. Some studies have highlighted the fact that HCC is actually a tumor type with low/moderate immunogenicity. Moreover, HBV and HCV, the biggest risk factors for HCC, have also been identified as promoting an immunosuppressive liver status and T cell exhaustion [28].

Based on the physiological characteristics of liver cancer, it causes serious risks related to hypervascularity and vascular abnormalities. This is why the target drug for liver cancer is focused on anti-VEGF (such as sorafenib/lenvatinib) [29]. Despite this, more effective molecules have not been found as therapeutic targets, which may be the main reason why these treatments only modestly improve survival. It is well established that cancer is an immune-related disease, and the advent of immunotherapy may be a turning point. The development of immune checkpoint inhibitors has transformed the treatment landscape for many malignancies because the VEGF pathway plays an important role in the immune-related microenvironment. Therefore, a treatment strategy combining the VEGF/PD-L1 blockade is being explored [30]. In 2020, a brand new therapy that attracted attention was finally introduced. The combination of atezolizumab (anti-PD-L1 immune checkpoint inhibitors) with bevacizumab (an anti-VEGF monoclonal antibody) is currently the first-choice first-line treatment, as it confers a superior survival benefit compared to sorafenib [13,31,32]. This treatment has been recommended and approved by many societies and countries as the first-line treatment for unresectable malignant liver cancer.

Drug combinations, including Tremelimumab (anticytotoxic T lymphocyte-associated antigen 4/anti-CTLA-4) and Durvalumab (anti-PD-L1), have been studied and discussed recently. They have shown favorable results in a phase II trial for unresectable liver cancer. This treatment strategy (STRIDE, single tremelimumab regular interval durvalumab) significantly enhanced overall survival compared to sorafenib [33]. With traditional treatments being ineffective for advanced liver cancer patients, the exploration of new methods is urgent. Nivolumab and ipilimumab (anti-CTLA-4 monoclonal antibody) have been approved by the FDA for second-line treatment of advanced HCC based on early efficacy and safety data [34,35].

Recently, various drugs have been developed for the treatment of HCC. Pembrolizumab (anti-PD-1 therapy) has shown significant improvements in overall survival, progression-free survival, and response rate for patients with advanced HCC [36]. Ramucirumab (anit-VEGFR-2) has been approved by the FDA as a second-line treatment for HCC in patients with an AFP ≥ 400 ng/mL after initial treatment with sorafenib [37]. There are still many different drugs that have been continuously explored and developed in clinical applications, but the study on drug resistance and biomarkers is relatively insufficient; thus, focus is on the discussion of the traditional drugs.

## 3. The Role of Biomarkers in Cancer Therapy

The emphasis of cancer treatment strategies has gradually shifted from surgery to chemical chemotherapy (e.g., doxorubicin) and finally to targeted therapy (e.g., TKI inhibitors, immune checkpoint inhibitors). Advances in cancer biology and sequencing technology have enabled the selection of targeted—and more effective—treatments for individual patients with various types of solid tumor [38,39]. Such evolving treatment strategies will very much depend on the patients themselves or the genes expressed by their tumor cells. Because biomarkers are critical to the rational development of individualized therapeutics (Figure 3) [40], such biomarker-based therapy can be expected to receive increasing attention. In a personalized medicine setting, biomarkers will be used not only to identify risks and monitor cancer, but also to select treatment methods. As the number of models designed to identify and predict biomarker genes capable of identifying drug susceptibility continues to grow, a priority for future research should be the development of new therapies targeting specific pathways involved in the pathogenesis of HCC [41,42,43]. Among the challenges in the clinical diagnosis of liver cancer is identifying biomarkers that satisfy the criteria of specificity, ease of detection (noninvasiveness), and the ability to reflect different pathological phenomena (tumor stage, size, invasion degree, and patient survival rate) [44]. Although still in the research stage, many biomarkers are available for use in evaluating the outcome of different treatments for liver cancer. In a clinical setting, potential targets for biomarker development include AFP-L3 (the Lens culinaris agglutinin-bound form of AFP), PIVKA-II (prothrombin induced by vitamin K absence-II), and DCP (Des-γ-Carboxy Prothrombin), for local treatments such as surgery and TACE; AFP, VEGF, and angiopoietin 2 (ANG-2) for systemic treatments such as TKIs; and inflammation-related factors such as PD1, PD-L1, and PD-L2 for immunotherapy [45].

The clinical application of biomarkers is not only to find “suitable” treatment strategies (PD-L1 for immunotherapy) but also to exclude “unsuitable” treatment options for patients. For example, in patients receiving immunotherapy, some people encounter immune-related adverse events, including fatigue, pruritus, rash, diarrhea, and increases in aspartate aminotransferase and alanine aminotransferase levels. Currently, elevated levels of serum IL-6 and IL-17 have been linked to a higher risk of severe side effects in patients receiving ipilimumab therapy, with IL-17 specifically linked to severe diabetes and colitis [42].

Moreover, genetic alterations also contribute to the ineffectiveness of treatment. Previously, studies have been conducted on patients with advanced liver cancer to analyze the genetic mutations of the disease and determine factors that predict resistance to systemic therapies using circulating tumor DNA (ctDNA). The most frequent mutations of advanced HCC were TERT promoter (51%), TP53 (32%), CTNNB1 (17%), PTEN (8%), AXIN1, ARID2, KMT2D, and TSC2 (each 6%) [46]. Moreover, some common mutations in HCC, such as in TERT, CTNNB1, and TP53 genes, are still considered to be undruggable [47]. Furthermore, some studies have indicated that the existence of primary and acquired drug resistance is the main impediment to the effect of the multikinase inhibitor sorafenib in advanced HCC. In addition, the discovery of a new somatic mutation in OCT4 (c.G52C) that affects the effect of sorafenib also provides new insights into the treatment of liver cancer and the selection of its treatment strategy based on the individual genetic patterns [48]. These clues indicate that in the application of biomarkers, in addition to the differences in gene expression, mutations are also a part that cannot be ignored.

## 4. Genes Associated with Doxorubicin Resistance

Doxorubicin, an antineoplastic agent in the anthracycline class, is a very commonly used chemotherapeutic drug in the clinical treatment of many malignant tumors, including liver cancer [49,50]. Among the general properties of drugs in this class are interaction with DNA through different mechanisms, including intercalation (insertion between base pairs), DNA strand breakage, and inhibition of the DNA-unwinding enzyme topoisomerase II. Doxorubicin also inhibits DNA polymerase activity, alters regulation of gene expression, and causes free radical damage to DNA [51]. Although considered a first-line drug in different types of cancers, doxorubicin faces two major obstacles in therapeutic applications: drug-induced cardiotoxicity and acquired drug resistance [52,53]. There are many mechanisms of drug resistance in cancer cells, including those involving the tumor suppressor p53, the multidrug resistance ABC transporter, and epithelial–mesenchymal transition (EMT), among others [54,55,56]. The mechanisms underlying these phenomena are always inseparable from the expression of specific genes. Therefore, an in-depth understanding of the differences in gene expression among different patients themselves ultimately informs choices regarding drugs used for treatment.

Expression of NAP1L1 (nucleosome assembly protein 1-like 1) is significantly upregulated in tumor tissues compared with adjacent nontumor tissues in many cancers, including liver, colon, ovarian, breast cancer, etc. [57,58,59,60]. High NAP1L1 expression in HCC tissues is associated with aggressive clinicopathologic features, such as high serum AFP levels, tumor size, and tumor number. It was also associated with poor overall survival in our cohort and an extravalidation cohort analyzed using a TCGA microarray dataset, and was further identified as an independent prognostic factor in HCC patients treated with radical resection. Similar to AFP, NAP1L1 is overexpressed in fetal liver compared with adult liver and is re-expressed in a subgroup of HCC patients with an unfavorable prognosis. Among the factors thought to be responsible for the development, recurrence, and chemoresistance of HCC are cancer stem cells (CSCs). In this context, it has been shown that expression of NAP1L1 is decreased concomitantly with the downregulation of certain CSC-related markers, including SOX2, OCT3/4, C-Myc, NOTCH1, and ABCG2. Importantly, NAP1L1 also inhibits the sensitivity of hepatoma cells to doxorubicin through its effect on CSCs [61]. Previous studies indicated that NAP1L1 downregulation may confer liver cancer cells’ sensitivity to doxorubicin by inhibiting the NOTCH1/ABCG2 signal pathway. NAP1L1 is a prognostic biomarker and contributes to doxorubicin chemotherapy resistance in human hepatocellular carcinoma [61]. In addition, some studies have reported that NAP1L1 is resistant to chemotherapy and radiotherapy in glioma [62,63].

CKLF1 (transcript variant 2 of the chemokine-like factor, CKLF) is a newly discovered CC chemokine that, together with STAT3, is elevated in HCC tissues compared with normal liver tissues, as determined from the Oncomine database using R software [64]. This study further showed that CKLF1 levels are correlated with vascular invasion and tumor size, and that overexpression of CKLF1 and STAT3 is associated with poor overall survival. Taken together with other findings reported in this study, these results indicate that CKLF1 might promote malignant transformation as well as invasion and metastasis in the development of HCC by activating the IL6/STAT3 signaling pathway. In many cancer cells, the IL-6/STAT3 pathway can cause drugs to lose their effect. Some studies have even pointed out that using drugs such as celecoxib to block the IL-6/STAT3 pathway can effectively resuscitate therapeutic strategies that have lost their efficacy owing to drug resistance [65]. Notably, CKLF1 acts through inhibition of the IL6/STAT3 pathway to prevent doxorubicin-induced apoptosis [64].

A trend in recent years in the selection of biomarkers has been a focus on nucleic acid candidates. Compared with protein biomarkers, nucleic acids offer advantages of high speed, low cost, ease of sampling, and high accuracy [44]. One such example is the long noncoding RNA, ARSR (lncARSR), which was shown to be significantly upregulated in HCC tissues compared with adjacent noncancerous liver tissues [66]. Additionally, several studies have found that lncARSR plays an oncogene role in various cancers, including renal cell carcinoma, colorectal cancer, bladder cancer, and non-small-cell lung cancer [67,68,69,70].

An analysis of the correlation between lncARSR expression and clinicopathological features showed that high expression of lncARSR was associated with liver cirrhosis, large tumor size, and advanced BCLC stage. Kaplan–Meier survival analyses further revealed that high expression of lncARSR is an indicator of poorer recurrence-free and overall survival. Mechanistically, lncARSR physically associates with PTEN (phosphatase and tensin homolog) mRNA and promotes its degradation, decreasing PTEN expression and activating the phosphoinositide 3-kinase (PI3K)/AKT pathway. This upregulation of the PTEN-PI3K/AKT pathway by elevated lncARSR expression underlies doxorubicin resistance in HCC, as evidenced by the fact that inhibition of the PI3K/AKT pathway abolishes lncARSR-dependent doxorubicin resistance in HCC cells, and depletion of PTEN reverses the effects of lncARSR knockdown on doxorubicin resistance in HCC cells [71]. It is important to note that lncARSR has been found to cause resistance to sunitinib and adriamycin in renal cell carcinoma and osteosarcoma [72,73].

In addition to the relatively novel lncRNAs, microRNAs (miRNAs) remain a focus of research on nucleic acid biomarkers [74]. miRNA-372-3p, which acts as a tumor suppressor in HCC, has been demonstrated to play a crucial role in cellular proliferation, apoptosis, and metastasis of various cancers, including HCC, colon cancer, and osteosarcoma [75,76,77]. Consistent with this, ectopic expression of miR-372-3p reduces cell proliferation and migration and significantly induces apoptosis in HepG2 cells in association with a decrease in the antiapoptotic protein Mcl-1 [75]. Notably, levels of miR-372-3p in the blood of HCC patients responsive to TACE with doxorubicin are significantly higher than those in nonresponders. In colorectal cancer, miR-372-3p interacts with LINC02418 to impact the effectiveness of 5-Fu chemoresistance [78].

## 5. Genes Associated with Capecitabine/5-FU Resistance

Capecitabine is an orally administered 5-FU prodrug that is absorbed in an intact form via the gastrointestinal tract. Capecitabine has been investigated in the treatment of advanced HCC using either convention [79].

CCND1 is the regulatory subunit of the holoenzyme that phosphorylates and inactivates retinoblastoma protein. Consistent with its function as an oncogene, CCND1 levels are elevated in multiple cancers, including HCC, and are associated with poor prognosis and tumor recurrence. CCND1 has been shown to participate in regulating DNA damage repair, which is associated with chemoresistance in cancer. In this context, silencing CCND1 suppresses DNA damage repair following 5-FU exposure, inhibiting the DNA repair protein RAD51 and enhancing levels of the DNA damage marker γ-H2AX [80]. In other cancers, CCND1 mutations promote ibrutinib resistance in mantle cell lymphoma [81].

Proteomic studies on the composition of a 5-FU–resistant hepatoma cell line have shown that the plasma membrane protein EPHX1 (epoxide hydrolase 1) is closely linked to 5-FU resistance and can be used as a new drug target to improve the clinical prognosis of patients with HCC [82]. Several studies have shown that low expression of EPHX1 is closely associated with different human cancers, including HCC, and that increased expression of EPHX1 suppresses cell proliferation, migration, and cycle progression, as well as inducing apoptosis in HCC [83]. The low-activity genotype of the EPHX1 gene is associated with decreased risk of lung cancer and head and neck cancer [84,85]. In acute myeloid leukemia, EPHX1 can affect the effect of chemotherapy by affecting drug metabolism and cell apoptosis [86].

Ubiquitin-specific protease 22 (USP22), which plays an important role in the resistance to chemotherapy drugs in HCC and the prognosis of treated patients by virtue of its function as a key subunit of the Spt-Ada-Gcn5 acetyl transferase complex, deubiquitinates the sirtuin family member, SIRT1 (silent information regulator 1). USP22 and SIRT1 are frequently highly expressed in HCC tissues. Overexpression of USP22 is associated with microscopic vascular invasion, but coexpression of USP22 and SIRT1 is more effective in predicting the prognosis of HCC. Inhibition of SIRT1 in vivo could be an effective strategy for improving 5-FU drug sensitivity, inhibiting tumor cell proliferation, and inducing apoptosis [87]. USP22 mediates the multidrug resistance of HCC via the SIRT1/AKT/MRP1 signaling pathway [88].

HCC tissues and cells express low levels of MiR-145, which is linked to high TNM staging and lymph node metastasis of HCC patients. Downregulation of miR-145 promotes drug resistance in HCC cells, inhibits apoptosis, and is an indicator of poorer prognosis. Consistent with this, overexpression of MiR-145 was shown to improve the sensitivity of HCC cells to 5-FU and enhance the inhibitory effect of 5-FU on tumor growth [89]. miR-145 has been recognized as a key suppressor of carcinogenesis and a factor of drug resistance in the lung, esophageal squamous cell carcinoma, glioma, breast cancer, and HCC [90].

High expression of miR-32-5p and low expression of its target gene PTEN are positively associated with poor prognosis. In accordance with this, miR-32-5p expression is significantly elevated in 5-FU-resistant HCC cells, whereas PTEN is reduced. This inverse correlation between miR-32-5p and PTEN has been confirmed in HCC cell lines and patients, and it has been found that miR-32-5p can also transfer to other liver cancer cells without 5-FU resistance through exosome, leading to an increase in miR 32-5p, a decrease in PTEN, and activation of the PI3K/Akt pathway, finally inducing 5-FU resistance [91]. However, instead, miR-32-5p could inhibit cellular malignant behavior by regulating the expression of HOXB8 in cervical cancer. The MiR-32-5p/HOXB8 axis might serve as a potential target for the clinical diagnosis and treatment of cervical cancer [92].

The involvement of a number of lncRNAs—some of which have been named—in resistance to 5-FU might provide insight into this inverse relationship. For example, the lncRNAs LINC01189, H19, and KRAL have been reported to act as tumor suppressors in liver cancer. Restoring their expression together with that of certain miRNAs (e.g., miR-155-5, miR-193a-3p, and miR-141) can restore sensitivity to 5-FU [93,94,95].

Other lncRNAs, such as LINC00680, which is more highly expressed in liver cancer tissues, can promote 5-FU resistance through effects on CSCs; specifically, upregulation of AKT3 by LINC00680 in these cells causes “sponging” of miR-568, thereby decreasing 5-FU chemosensitivity [96].

Recent bioinformatics analyses of the TCGA LIHC dataset identified the lncRNA PLEKHA8P1 and its parental gene PLEKHA8, a well-studied transport protein in the Golgi complex, as oncogenes in both colorectal and liver cancer. This analysis further revealed that this gene pair promotes tumor progression and that their dysregulated expression affects resistance to 5-FU in human HCC cells [97].

## 6. Genes Associated with Sorafenib Resistance

Sorafenib is a multikinase inhibitor capable of facilitating apoptosis, mitigating angiogenesis, and suppressing tumor cell proliferation. Although sorafenib is currently an effective first-line therapy in late-stage HCC, it is unfortunately associated with the increasingly common problem of acquired drug resistance [98]. Regardless of the reason, drug resistance is inseparable from genetic changes.

Cell surface profiling is of special interest in oncoproteomics as it may provide more accurate and reliable diagnosis and prognosis and identify new therapeutic targets [99]. It is also an important tool for assessing drug resistance. The surface protein CD24, which has been implicated in sorafenib resistance, is overexpressed in HCC tumor tissues and in highly metastatic HCC cell lines. Plasma CD24 levels are significantly higher in HCC patients than in controls and are associated with tumor differentiation. Those HCC patients with high plasma CD24 levels have been shown to have an unfavorable prognosis [100]. Notably, there is a positive correlation between CD24 expression levels and sorafenib resistance. CD24 leads to an increase in PP2A protein production and causes deactivation of the mTOR/AKT pathway; this, in turn, enhances autophagy activity, which is accompanied by CD24-related sorafenib resistance. The combination of autophagy modulation and CD24 targeted therapy is a promising strategy in the treatment of HCC [101]. CD24 expression is also associated with the development, invasion, and metastasis of ovarian cancer. CD24 has been identified as an independent prognostic marker of survival in patients with ovarian cancer [102]. CD24 plays an important role in the development of anoikis resistance and CD24 can be used as a new therapeutic target to induce anoikis and inhibit metastasis in ovarian cancer [103].

High expression of the surface protein CD13 in CSCs is correlated with early, vimentin-associated recurrence and poor prognosis in HCC. CD13 represents a potential therapeutic target in the treatment of HCC because regulation of CSCs and suppression of EMT are essential in cancer therapy [104]. CD13 promotes HCC proliferation, invasion, cell cycle progression as well as sorafenib resistance. CSCs are naturally resistant to chemotherapy, which is one reason why tumors often recur after initial remission following chemotherapy [105]. In HCC, CD13 interacts with HDAC5 (histone deacetylase 5) to promote its protein stability, thus resulting in HDAC5-mediated LSD1 (lysine-specific demethylase 1) deacetylation and protein stabilization. LSD1 decreases the NF-κB catalytic unit p65 methylation that leads to p65 protein stability. NF-κB activation leads to gene activation involved in the cell cycle and antiapoptosis, including—but not limited to—Cyclin A, BCL2, MCL-1, and BCL-xL. The central role of the CD13-HDAC5-LSD1-NF-κB signaling axis is its contribution to HCC tumorigenesis and sorafenib resistance [106]. Moreover, CD13 promotes tumor angiogenesis, invasion, and metastasis in breast, ovarian, and prostate cancer cells by participating in enzymatic cleavage of the polypeptide chain. More importantly, CD13 is also associated with the multidrug resistance (MDR) of these tumors, increasing various drug resistances [107].

In addition, there are a number of protein-coding genes related to resistance to sorafenib, many of which have extremely high clinical potential against liver cancer. Peroxisome proliferator-activated receptor gamma coactivator-1β (PGC-1β), a transcriptional activator of medium-chain acyl-CoA dehydrogenase (MCAD) and long-chain acyl-CoA dehydrogenase (LCAD), enhances expression levels of both MCAD and LCAD, which are key enzymes necessary for fatty acid oxidation, and thus are critical for promoting tumorigenesis [108]. UBQLN1, a member of the ubiquilin (UBQLN) family of ubiquitin-like proteins, plays an important role in maintaining protein homeostasis through regulation of the ubiquitin–proteasome system (UPS), macroautophagy (autophagy), and ER-associated protein degradation (ERAD). It also participates in the regulation of a variety of physiological and pathophysiological phenomena [109]. Mechanistically, sorafenib targets electron transport chain complexes, resulting in the generation of reactive oxygen species (ROS). However, studies have shown that cancer cells develop defense mechanisms that result in resistance to sorafenib-generated ROS. In HCC, upregulated UBQLN1 induces degradation of PGC-1β in a ubiquitination-independent manner to attenuate mitochondrial biogenesis and ROS production in cells with acquired resistance to sorafenib [110].

Recent evidence suggests an association between CXCL/CXCR chemokines and chemotherapy resistance in cancer cells. The association between metabolic alterations and sorafenib resistance in HCC cells, together with the demonstrated importance of CXCR3 in the development of resistance to sorafenib in HCC cells and the novel mechanism of CXCR3 regulation of the AMPK pathway activity and adipocytokine signaling, suggest that lipid peroxidation results in metabolic alterations during chemoresistance [111]. In addition, CXCR3 has been implicated in causing chemoresistance and immunotherapy resistance by affecting stem-like ability or immune function in head and neck cancer and breast cancer [112,113].

Eph receptor B2 (EphB2), an important member of the Eph receptor family, has been demonstrated to be aberrantly expressed in many cancer types, including colorectal cancer, gastric cancer, and HCC, where it results in tumor occurrence and progression [114]. An RNA-sequencing analysis of a sorafenib-resistant model identified EPHB2 as the most significantly upregulated kinase, and further showed that EPHB2 expression increased stepwise from normal liver tissue to fibrotic liver tissue to HCC tissue and was correlated with poor prognosis. Mechanistically, EPHB2 regulates cancer stemness and drug resistance by driving the SRC/AKT/GSK3β/β-catenin signaling cascade. EPHB2 expression, in turn, is transcriptionally regulated via its promoter by TCF1, forming a positive Wnt/β-catenin feedback loop. Targeting such a positive feedback loop involving the EPHB2/β-catenin axis may be a possible therapeutic strategy for combating acquired drug resistance in HCC [115].

Increasing evidence demonstrates that RNA modifications play an important role in the progression of HCC. The m7G methyltransferase WDR4 (WD repeat domain 4) is upregulated in HCC and promotes cell proliferation by inducing G2/M cell cycle transition and inhibiting apoptosis, which enhances metastasis; it also enhances sorafenib resistance through EMT. Furthermore, WDR4 transcription can be induced by c-Myc and the resulting increase in WDR4 protein promotes CCNB1 mRNA stability and translation, thereby enhancing HCC progression. In terms of mechanism of action, WDR4 enhances CCNB1 translation by promoting the binding of EIF2A to CCNB1. Increased levels of CCNB1 protein, in turn, activate the PI3K/AKT pathway in HCC, subsequently reducing P53 protein expression by promoting P53 ubiquitination. Thus, the m7G methyltransferase WDR4 is a tumor promoter in the development and progression of HCC and may be a candidate therapeutic target in HCC treatment [116]. A recent report indicated that WDR4 plays a significant role in various kinds of malignant tumors for prognostic prediction and carcinoma drug resistance prediction after analyzing WDR4 through human pan-cancer [117].

MiRNAs are of considerable importance in cancer biology and potential clinical applications, and may also play an important role in sorafenib resistance. MiR-378a-3p expression is significantly downregulated in HCC, and this decreased expression is associated with higher microvascular density. MiR-378a-3p downregulation is an indicator of shortened survival time in HCC patients [118]. In addition, it has been shown that miR-378a-3p levels exhibit a downward trend in HCC and are negatively correlated with PD-L1 levels, indicating that miR-378a-3p also plays an important predictive role in immunotherapy [119]. MiR-378a-3p expression is frequently reduced in established sorafenib-resistant HCC cell lines, and decreased miR-378a-3p levels correlate with poor overall survival of HCC patients following sorafenib treatment [120]. Furthermore, miR-378a-3p promotes breast cancer stemness and doxorubicin and paclitaxel resistance via the activation of EZH2/STAT3 signaling [121]. On the contrary, miR-378a-3p enhanced the sensitivity of ovarian cancer cells to cisplatin through targeting MAPK1 and GRB2 [122].

MiR-30a-5p plays vital roles in the carcinogenesis and progression of various malignancies via different molecular mechanisms [123]. miR-30a-5p expression is significantly lower in HCC tumor tissue compared with adjacent normal tissue and is correlated with tumor nodes, metastasis, the tumor–node–metastasis stage, portal vein tumor embolus, vascular invasion, and tumor capsule status [124]. The aberrant glucose metabolism in cancer cells, reflecting a shift away from oxidative phosphorylation towards aerobic glycolysis during tumorigenesis (i.e., Warburg effect), is associated with resistance to chemotherapeutic agents. The central role of CLCF1 in promoting glycolysis through activation of PI3K/AKT signaling and its downstream genes thus participates in glycolysis in sorafenib-resistant HCC cells. Sorafenib-mediated suppression of miR-30a-5p, which acts as a transcriptional repressor of CLCF1, results in upregulation of CLCF1 in sorafenib-resistant HCC cells [105]. MiR-30a-5p is expressed in in various types of cancer, including lung, ovarian, and gastric cancer. It affects the treatment efficacy by regulating the expression of such as CHD1, YAP1, and ROR1 [125,126,127].

MiR-21 has been demonstrated to play a role in many types of cancer and has been shown to be associated with HCC prognosis in various studies. MiR-21 expression is significantly upregulated in HCC tissues and is an independent prognostic factor for overall survival and disease-free survival in HCC patients who undergo surgical resection [106]. Moreover, the AKT pathway is highly activated by overexpressed miR 21 in sorafenib-resistant HCC cells compared with parental HCC cells [128]. As well as in liver cancer, miR-21 is an authentic oncogene in mediating drug resistance in several other cancers, including breast cancer and gastric cancer [129,130].

In terms of noncoding genes, many lncRNAs also play important roles in sorafenib resistance in addition to miRNAs. Research on lncRNAs has demonstrated that a large part of the contribution of lncRNAs is attributable to their relationship with miRNAs. As mentioned above, miR-21 can modulate AKT effects and lead to sorafenib resistance; a subsequent study found that this mechanism is also related to the lncRNA SNHG1 (small nucleolar RNA host gene 1) [128]. SNHG1 has been shown to play roles in the development and progression of various tumors, including HCC, breast, and gastric cancer.

SNHG1 upregulation is associated with the promotion of several primary biological functions, including cell proliferation, transcription, and protein binding [110,131,132,133].

Upregulation of the lncRNA TRERNA1 (translation regulatory lncRNA 1) by hepatitis B virus-encoded X (HBx) protein not only promotes cell proliferation but is also positively correlated with poor prognosis in HCC. Importantly, TRERNA1 was shown to enhance sorafenib resistance in HCC cells. Mechanistically, TRERNA1 acts as a competitive endogenous RNA (ceRNA) that regulates NRAS expression by acting as a sponge for miR-22-3p, thereby activating the RAS/RAF/MEK/ERK signaling pathway [134].

The lncRNA HOTAIR (HOX transcript antisense intergenic RNA) has been classified as an oncogene that accelerates cell proliferation, migration, and invasion in many cancer types by interacting with miRNAs [116]. It has been reported that sorafenib resistance is increased in HCC cells with high HOTAIR expression and that HOTAIR increases sorafenib resistance in HCC by inhibiting miR-217 [135]. In addition to liver cancer, HOTAIR has been shown to play a role in promoting the proliferation, survival, invasion, metastasis, and gefitinib resistance in lung cancer [136,137].

Expression of the lncRNA NEAT1 (nuclear enriched abundant transcript 1) was shown to be significantly increased in HCC tissues compared with adjacent tissues and negatively correlated with survival of HCC patients. Downregulation of NEAT1 inhibits EGFR expression, promotes hepatoma cell apoptosis, and inhibits the cell cycle, thereby inhibiting tumor proliferation and invasion [138]. NEAT1 expression was shown to be significantly associated with HCC prognosis and sorafenib resistance patterns. Like other lncRNAs, NEAT1 acts a sponge for miRNAs, in this case miR-204; the upregulation of ATG3 (autophagy related 3) expression associated with decreased levels of miR-204 leads to enhanced cellular resistance to sorafenib [139]. NEAT1 also targets miR-149-5p and thereby decreases the activity of sorafenib against HCC cells. It has been demonstrated that NEAT1 functions are triggered by regulation of the miR-149-5p/AKT1 axis [140]. NEAT1 also promotes the paclitaxel resistance of breast cancer via sponging the miR-23a-3p-FOXA1 axis and promotes the cisplatin resistance in ovarian cancer by regulating the miR-770-5p/PARP1 axis [141,142].

## 7. Genes Associated with Lenvatinib Resistance

Lenvatinib, a multitargeted inhibitor that suppresses VEGFR, FGFR, and PDGFRα as well as the proto-oncogenes RET and KIT, is an emerging first-line therapy for HCC. With the approval of lenvatinib, sorafenib is no longer the only TKI inhibitor option for liver cancer treatment [20]. Preclinical studies have shown that lenvatinib has potent antiangiogenic activity, mainly through inhibition of the VEGF and FGF signaling pathways [143,144]. Acquired resistance to lenvatinib in the context of treatment of liver cancer, a clinical experience similar to that encountered with sorafenib, has received considerable research attention. This problem was recently addressed using CRISPR/Cas9 library screening, which identified two key resistance genes, neurofibromin 1(NF1) and dual-specificity phosphatase 9 (DUSP9), as critical drivers of lenvatinib resistance in HCC [145]. Neurofibromin products of the neurofibromatosis type 1 (NF1) gene play an important role in tumor suppression [146]. RNA interference (RNAi)- and CRISPR/Cas9-based knockout models have shown that loss of NF1 reactivates PI3K/AKT and MAPK/ERK signaling pathways and induces lenvatinib resistance [145]. Unfortunately, however, there is limited research on the role of NF1 in liver cells, so it may be premature to use NF1 as a biomarker for liver cancer treatment. A similar situation applies to DUSP9, which is equally difficult to apply clinically, owing to the few related studies and lack of clarity regarding its role in the disease [147,148].

Activation of the HGF/c-Met axis, which is associated with tumor progression and several hallmarks of cancer, is considered a key contributor to drug resistance. HGF (hepatocyte growth factor) reduces the antiproliferative, proapoptotic, and anti-invasive effects of lenvatinib on HCC cells with high expression of the RTK, c-Met, but does not significantly affect HCC cells with low c-Met expression. Moreover, combining lenvatinib treatment with a c-Met inhibitor may improve its systemic therapeutic efficacy in HCC patients with high c-Met expression [149].

As noted above, CSCs are positively associated with resistance to various drugs, including lenvatinib, and this resistance is also affected by genes related to the functional properties of CSCs. As an example, expression of CD73 is positively associated with sphere-forming capacity and is elevated in HCC spheroids. Consistent with this, knockdown of CD73 attenuates sphere formation, lenvatinib resistance, and stemness-associated gene expression, whereas CD73 overexpression causes the opposite effects [150]. The elevated expression levels of exosomal CD73 affect the response to anti-PD-1 agents in patients with melanoma who failed to respond to therapy. This is because CD73 expressed on exosomes from the serum of patients with melanoma produces adenosine and contributes to suppressing T-cell functions [151].

CD133 is one of the best-characterized markers of CSCs in various tumor types, including HCC. It has been demonstrated that CD133-positive cells are involved in metastasis, tumorigenesis, tumor recurrence, and resistance to treatment in HCC. The potential of CD133 for use in clinical prognosis prediction and targeted therapy underscores the clinical significance of CD133 in HCC [152]. Hedgehog signaling is also important for the maintenance of HCC stemness. Notably, the Hedgehog signaling inhibitor, GANT61, was shown to reverse lenvatinib resistance by suppressing Hedgehog signaling in HCC cells, especially in CD133-positive cells. Moreover, combining lenvatinib with Hedgehog signaling inhibitors was found to improve its therapeutic efficacy in HCC patients with high CD133 expression levels [153].

Interferon regulatory factor 2 (IRF2) is a constitutive transcription factor associated with the development of various cancers through the regulation of cancer cell growth, apoptosis, and drug resistance. Specifically, IRF2 promotes proliferation, inhibits apoptosis, and increases resistance of HCC cells to lenvatinib by regulating β-catenin expression. Moreover, inhibiting β-catenin with XAV-939 was shown to effectively abrogate lenvatinib treatment-induced β-catenin expression [154].

A comprehensive RNA sequencing (RNA-seq) analysis of lenvatinib-resistant HCC cell lines for factors that critically contribute to lenvatinib resistance identified integrin subunit beta 8 (ITGB8) as a crucial driver of lenvatinib resistance in HCC cells. Mechanistically, ITGB8 was shown to regulate lenvatinib resistance through HSP90-mediated stabilization of AKT and enhanced AKT signaling [89]. In lung cancer, circDNER enhances paclitaxel resistance and tumorigenicity of lung cancer via targeting miR-139-5p/ITGB8 [155]. Moreover, miR-199a-3p also enhances the cisplatin sensitivity of ovarian cancer through targeting ITGB8 [156].

YRDC, an N6-threonylcarbamoyltransferase domain-containing protein involved in cell growth, telomere homeostasis, translation, and N6-threonylcarbamoylation (t6A) of tRNA, was shown to be abnormally expressed during tumor progression. Not only is YRDC expression dysregulated in HCC tissue, its expression is related to the prognosis of HCC patients. Specifically, YRDC promotes the progression of HCC by activating the MEK/ERK signaling pathways [157]. It also promotes the proliferation of HCC cells by regulating the activity of the RAS pathway—the primary pathway impacted by the anticancer effects of lenvatinib. Notably, YRDC was found to mediate the resistance of lenvatinib in HCC cells by modulating translation of the proto-oncogene K-Ras [158]. Consistent with this, YRDC has been confirmed as a potential predictive biomarker of lenvatinib sensitivity in HCC. Moreover in NSCLC, RNA structural dynamics modulate EGFR-TKI resistance through YRDC [159].

Mucin 15 (MUC15) has also been linked to lenvatinib resistance. MUC15 inhibits hepatoma cell self-renewal, malignant proliferation, tumorigenicity, and chemoresistance by interacting with c-Met and subsequently inactivating the PI3K/AKT/SOX2 signaling pathway. The MUC15/c-Met/PI3K/AKT/SOX2 axis determines the responses of hepatoma cells to lenvatinib treatment, as supported by the observation that MUC15 overexpression abrogates lenvatinib resistance [160]. In addition, miR-552 promotes the proliferation and metastasis of cervical cancer cells by targeting the MUC15 pathway [161].

We previously noted the important influence of c-Met on lenvatinib resistance, commenting on the fact that miR-128-3p, which is downregulated in lenvatinib-resistant HCC cells, exerts the most vigorous negative regulation of c-Met [162]. MiR-128-3p is frequently downregulated in HCC tissues and suppresses HCC cell proliferation by regulating PIK3R1 (phosphoinositide-3-kinase regulatory subunit 1), suggesting that miR-128-3p is a potential indicator of the prognosis of HCC patients [163,164]. Additionally, miR-128-3p has been shown to influence the chemosensitivity of oxaliplatin resistance in colorectal cancer [165].

## 8. Genes Associated with Nivolumab and Pembrolizumab Resistance

Resistance to immunotherapy differs from that to chemotherapy and targeted therapy as it is mainly closely related to the patient’s own immune function and tumor type [166]. Moreover, only a minority of patients are expected to experience a positive response to PD1/PD-L1 blockade therapy, and primary or acquired resistance might eventually lead to cancer progression in patients with clinical responses [167]. Therefore, most biomarker studies on immunotherapy have focused on PD1/PD-L1 and genes whose products affect the immune system. In addition to the expression of PD1 and PD-L1, which are biomarkers of inflammation, inflammatory genes including CD274, CD8A, LAG3, and STAT1 have been linked to improved clinical outcomes, including enhanced responsiveness and survival [168].

The proinflammatory Th1 cytokines, tumor necrosis factor (TNF)-α, and interferon (IFN)-γ, are indispensable coordinators of the inflammatory responses involved in HBV pathogenesis. TNF-α and IFN-γ levels are markedly increased during HCC development [169]. The transcription factors interferon regulatory factor 1 (IRF-1) and IRF-2 regulate PD-L1 in HCC cells. IFN-γ induces PD-L1 mRNA and protein expression through upregulation of IRF-1 in mouse and human HCC cells, and it has been shown that IRF-1 mRNA expression is increased in patients with well-differentiated or early-stage HCC tumors. IRF-1 has traditionally been viewed as a tumor suppressor. However, these novel findings reveal a complex role for IRF-1, which upregulates PD-L1 in the inflammatory tumor microenvironment. Specifically, they show that IRF-1 antagonizes IRF-2 for binding to the IRE promoter element in PD-L1, providing new insight into the regulation of PD1/PD-L1 pathways in HCC immune checkpoint blockade therapy [170]. Thus, the study of these critical immune and inflammatory genes is important not only in liver cancer but also in other cancer types, and the detection of biomarkers associated with these genes can aid in treatment decision making for a wide range of cancer patients. [171].

Pyruvate kinase M2 (PKM2), an essential regulator of the Warburg effect responsible for the metabolic shift in cancer cells, is significantly upregulated in HCC. PKM2 is further associated with poor prognosis in HCC patients [172]. It is also strongly correlated with the expression of immune inhibitory cytokines and lymphocyte infiltration in HCC. Overexpression of PKM2 was shown to sensitize HCC to the immune checkpoint blockade, enhancing IFN-γ–positive CD8 T cells. PKM2 might have prognostic value and could be a potential therapeutic target for immune checkpoint inhibitors in HCC [173]. Furthermore, cisplatin-resistant NSCLC cells, also induced by hypoxia, transmit resistance to sensitive cells through exosomal PKM2 [174]. Similarly, the ubiquitination of PKM2 also plays an important role in the doxorubicin resistance of breast cancer [175].

Natural killer (NK) cells play a critical role in the innate antitumor immune response, and dysfunction of NK cells has been verified in various malignant tumors, including HCC. Increased levels of circUHRF1 (circular ubiquitin-like with PHD and ring finger domain 1 RNA) are indicative of NK cell dysfunction and poor clinical prognosis in HCC patients. Notably, the expression of circUHRF1 is higher in human HCC tissues than in matched adjacent normal tissues. circUHRF1 in HCC patient plasma is predominantly secreted by HCC cells through exosomes and inhibits NK cell-mediated secretion of IFN-γ and TNF-α. A high level of plasma exosomal circUHRF1 is associated with a decreased proportion of NK cells and decreased NK cell tumor infiltration. CircUHRF1 may drive resistance to anti-PD1 immunotherapy, providing a potential therapeutic strategy for patients with HCC [176].

## 9. Discussion

Biomarkers are widely used in the clinical management of liver cancer, from assessing the initial cancer risk and midterm treatment strategies to monitoring final prognosis. This invaluable versatility compels us to pay greater attention to and invest more research on biomarkers. In this review article, we have sought to present an integrated view of biomarkers that may be helpful in clinical treatment of liver cancer, particularly as it relates to identifying candidate biomarkers related to resistance against different drugs used in the treatment of liver cancer. Returning to questions related to the application of biomarkers, we reiterate the importance of the key characteristics of specificity, ease of detection, and stability. For example, in liver cancer, the most widely used biomarker, AFP, suffers from poor specificity; thus, more research is needed to find alternative biomarkers. As for ease of detection, the focus is on liver cancer cells, where increases in expression level occur and where biomarkers can be detected mainly through noninvasive approaches. Previous studies have noted that proteins are not the only suitable biomarkers; in fact, nucleic acid-type biomarkers offer greater advantages compared with protein-based options, such as lower cost and richer and faster detection technology, among others. Therefore, in discussing biomarker selection, we give priority to nucleic acid biomarkers. For example, in doxorubicin-based strategies for the treatment of liver cancer, lncARSR is among the biomarker candidates that can be detected. Since lncARSR itself is a nucleic acid localized to exosomes, detection is simple and largely noninvasive using the patient’s blood or urine, thereby reducing patient discomfort [177]. In terms of pathological phenomena, the expression of lncARSR can reliably reflect the course of liver cancer through its different stages [71]. Importantly, all current studies on lncARSR in the treatment of liver cancer resistance have focused on doxorubicin, which means that the detection of lncARSR is highly specific and can be used to identify whether patients are suitable for treatment with doxorubicin [178]. Of course, there are also studies that have found that lncARSR is related to the function of CSCs [66]. Therefore, additional research is needed to confirm the specificity of lncARSR in the resistance to doxorubicin.

Similarly, in selecting treatment regimens using capecitabine/5-FU, detection of miR-32-5p can also be considered a suitable biomarker strategy. Although miR-32-5p has been linked to many different cancers, where it is discussed in the context of an oncogene, in liver cancer, research on the relationship between miR-32-5p and drug resistance still only applies to capecitabine/5-FU and related compounds. On the other hand, the fact that miR-32-5p is a resident of exosomes also aids the stability of this miRNA in the blood and facilitates its easy detection [91,92,179]. Notably, the lncRNA LINC00680 shares the same potential as miR-32-5p [96].

Given the importance of sorafenib in liver cancer therapy and treatment failures associated with the development of severe resistance to it, it is important to effectively identify biomarkers of resistance in patients. miR-30a-5p is a very good choice for this purpose. In addition to the fact that miR-30a-5p contributes to sorafenib resistance in liver cancer patients in several different ways [180,181], it also has the potential for noninvasive diagnosis and reflects the pathological phenomena of liver cancer at different stages, including survival rate [124,182]. Unfortunately, previous studies have highlighted the role of miR-30a-5p as a tumor suppressor, which means that it may be difficult to detect at its lower expression levels [123,183]. In addition, although it meets our expectations as a biomarker in that it is highly expressed in liver cancer patients and can respond to sorafenib resistance properties, miR-21 has been reported in many studies on resistance related to different anticancer drugs, including cisplatin, 5-FU, and doxorubicin [184,185,186]. Therefore, miR-21 lacks the specificity to guide selection of a suitable drug for clinical application.

In contrast, the lncRNA SNHG1 has a certain potential for use in the development of liver cancer therapy as a biomarker of sorafenib resistance [128]. Although the SNHG1 gene plays an important oncogenic role in liver cancer, this is also the case for most cancers, including colorectal, cervical, and colon cancers [187,188,189]. However, for the purpose of selecting treatment methods, the nonspecificity of the cancer type is not a primary consideration in biomarker detection compared with the nonspecificity of anticancer drug resistance for patients who have been clinically diagnosed with liver cancer. In this context, the research on SNHG1 to date only relates to sorafenib resistance, so it is thought that it has the potential for further development. The lncRNA TRERNA1, which we did not include in the discussion, is also a nucleic acid that is only associated with sorafenib resistance [134], but it lacks SNHG1′s advantage of being readily detected by virtue of its presence in the exosome [185]. For reasons similar to those that apply to SNHG1, the lncRNA NEAT1 possesses properties that make it very suitable for further exploration [139,190,191].

Lenvatinib is a targeted-therapy drug that is very similar to sorafenib in terms of mechanism of action and clinical efficacy, so specificity may be the most important criteria in the selection of suitable biomarkers. However, most research on lenvatinib resistance has focused on coding genes, so finding support in the literature for the selection of nucleic acids as candidate biomarkers can be difficult. In terms of nucleic acid options, the lncRNA MT1JP is upregulated in lenvatinib-resistant HCC cells and inhibits apoptosis signaling [192]. However, the selection of MT1JP as a biomarker in liver cancer is complicated by its apparent dual roles: in addition to being associated with lenvatinib resistance owing to its antiapoptotic ability, some studies have noted that MT1JP is actually a tumor suppressor [193,194]. Given the complexities involved in choosing and detecting a biomarker, additional studies will be required to determine the clinical value of MT1JP. Unfortunately, at this stage, there is a relative paucity of nucleic acid-related biomarker studies on lenvatinib resistance compared with other therapeutic drugs, making it relatively difficult to find meaningful and specific candidate genes for further consideration.

Nivolumab and pembrolizumab are two recently developed anticheckpoint monoclonal antibodies that target PD1 [27]. Although both agents work according to the same basic principle, their effects are slightly different, although not necessarily uniform, across different cancers. For example, some studies have noted that the clinical efficacy of pembrolizumab in liver cancer is greater than that of nivolumab, but in non-small-cell lung cancer, there is no difference between the two [195,196]. As for resistance to the two drugs in liver cancer, our view is that it is generally directly related to the patient’s own immune function. Because of the involvement of many inflammation-related factors, it remains difficult to find and implement reliable biomarkers capable of predicting resistance to immunotherapy drugs.

In summary, drug resistance in liver cancer and other cancers is a complex and multifactorial phenomenon. While several genes have been identified to contribute to drug resistance, a strong target gene for drug development is still lacking in the therapeutic strategy of liver cancer. The current mixed therapy (such as atezolizumab plus bevacizumab) seems to be promising, but it needs more time to be tested. Additionally, there are two types of drug resistance that appear during cancer treatment: inherent and acquired drug resistance, and biomarkers can be used to identify these genes before and during treatment. Further research is needed to discover a key gene in the clinical treatment of liver cancer and as a biomarker for detection and evaluation.

## 10. Conclusions

In conclusion, although our discussion has focused on nucleic acid biomarkers, there is no denying the research value of protein biomarkers. Purely from a technological perspective, whether next-generation sequencing or RNA sequencing, the rapid development of biotechnology in recent years has tended to favor nucleic acids. This trend should ultimately lead to greater discussion of nucleic acid-type biomarkers with more clinical advantages. However, despite the many potential biomarkers identified during the course of research on various therapeutic drugs under development, to date, these candidates remain far from actual clinical application. Hopefully, future research along these lines will ultimately lead to a breakthrough in the monitoring and treatment of patients with liver cancer.

## Figures and Tables

**Figure 1 cells-12-00869-f001:**
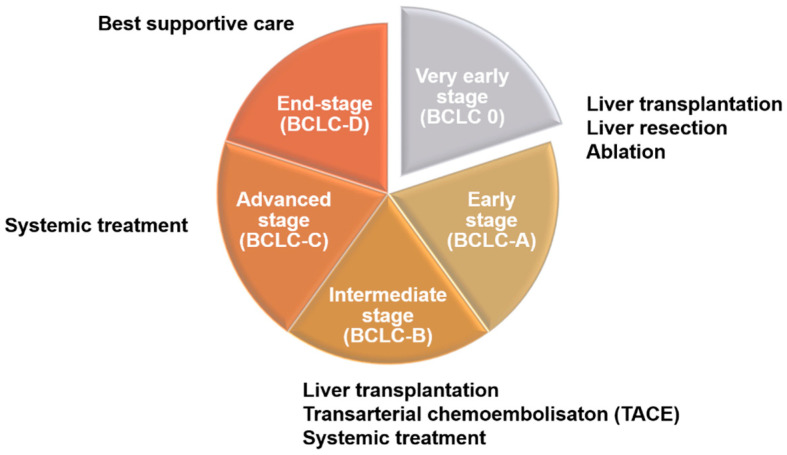
Treatment strategies for liver cancer at different stages. Patients in the early stage of liver cancer (0-A) are generally treated with surgery, such as liver resection, liver transplantation, and ablation. Intermediate-stage patients (B), after evaluation, will begin to receive systemic chemotherapy or transarterial chemoembolization (TACE). In the advanced stage, patients can be treated with second-line drugs that are different from first-line drugs after evaluation. At the end stage, if the treatment of patients has not significantly improved the survival rate, best supportive care should be used.

**Figure 2 cells-12-00869-f002:**
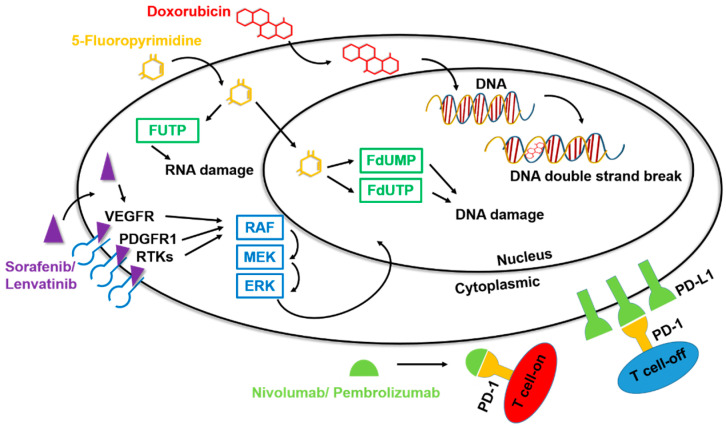
Mechanisms of action of drugs for HCC. Among the drugs for the clinical treatment of liver cancer, chemotherapeutic agents such as doxorubicin and 5-fluoropyrimidine can cause formation of unstable DNA in cells through the chemical structure, promoting cell apoptosis and inhibiting the carcinogenic process. The targeted drugs sorafenib and lenvatinib inhibit multiple tyrosine kinases, specifically affecting certain target RTKs such as VEGFR and PDGFR, thereby impacting downstream signal transmission. The immunotherapy drugs nivolumab and pembrolizumab have an anti-PD1 function, binding to T cells and preventing PD1 binding, thereby disrupting the inhibitory action of T cells.

**Figure 3 cells-12-00869-f003:**
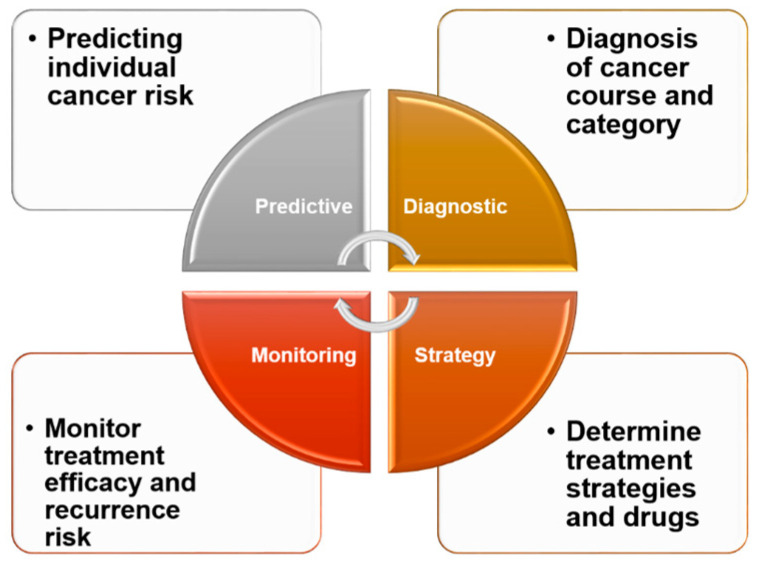
Clinical significance of biomarkers. In clinical practice, biomarkers can be effective and convenient for healthy people, providing insight into their own conditions and the risk of developing diseases. For cancer patients, biomarkers can be helpful in understanding their type of cancer and various tumor processes. From the standpoint of treatment, biomarkers can be an effective tool to aid patients and their doctors in selecting from among different treatment options to reduce drug resistance and increase treatment efficacy. Regular monitoring of various biomarkers in patients receiving treatment can also help patients to understand their own recovery and assess the likelihood of recurrence.

## Data Availability

Not applicable.

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
