# Peer review of "Evaluation and Application of Drug Resistance by Biomarkers in the Clinical Treatment of Liver Cancer"

_cells, 2023, doi:10.3390/cells12060869_

Round 1

Reviewer 1 Report

1.        The Figure 1 is incorrect. In early stage HCC, RFA is also the treatment of choice in addition to surgical resection. In intermediate HCC, chemotherapy does not play a role. Systemic therapies (including targeted therapy and immunotherapy) are both the choices for BCLC stage C patients. While BCLC stage D is terminal stage. It is weird that the authors classified late stage into late I and late II, which is not mentioned in any guideline. TACE is classified as local therapy, not chemotherapy, in the treatment of HCC.

2.        The description in the “2. Clinical treatment strategies for liver cancer” is out-of-date. Please refer to the newest BCLC guideline, published in 2022. The whole paragraph should be re-written.

3.        The clinical data on targeted therapy or immunotherapy are out-of-date. A lot of real-world experiences have been published in 2021 and 2022. I would encourage the authors to update the clinical data.

4.        The authors did not mention atezolizumab+bevacizumab (A+B) for HCC (line 130-138). A+B has been recommended as the first-line therapy for unresectable HCC in many guidelines (EASL, AASLD, ESMO etc).

5.        The authors did an extensive review on the biomarkers related to drug resistance in each drug. However, it would be better for the authors to provide insights on this issue in the Discussion section.

Author Response

Open Review 1

  1. The Figure 1 is incorrect. In early stage HCC, RFA is also the treatment of choice in addition to surgical resection. In intermediate HCC, chemotherapy does not play a role. Systemic therapies (including targeted therapy and immunotherapy) are both the choices for BCLC stage C patients. While BCLC stage D is terminal stage. It is weird that the authors classified late stage into late I and late II, which is not mentioned in any guideline. TACE is classified as local therapy, not chemotherapy, in the treatment of HCC.

We have drawn a new Figure 1 and rewritten this paragraph with reference to the BCLC guidelines issued in 2022.

Page.3, lines.90-96

  1. The description in the “2. Clinical treatment strategies for liver cancer” is out-of-date. Please refer to the newest BCLC guideline, published in 2022. The whole paragraph should be re-written.

We have rewritten this paragraph.

Page.2, lines.79-88

  1. The clinical data on targeted therapy or immunotherapy are out-of-date. A lot of real-world experiences have been published in 2021 and 2022. I would encourage the authors to update the clinical data.

We have added more new content.

Page.5, lines.191-207

  1. The authors did not mention atezolizumab+bevacizumab (A+B) for HCC (line 130-138). A+B has been recommended as the first-line therapy for unresectable HCC in many guidelines (EASL, AASLD, ESMO etc).

We have added relevant content.

Page.5, lines.176-190

  1. The authors did an extensive review on the biomarkers related to drug resistance in each drug. However, it would be better for the authors to provide insights on this issue in the Discussion section.

We have added more insights to the Discussion.

Page.17, line.795-803

  1. The manuscript has been revised by a native English speaker.

Reviewer 2 Report

In this manuscript, Huang et al. describe that biomarkers for drug resistance in liver cancer therapy. The authors reviewed several articles regarding the clinical treatment of liver cancer and described the genes which have the potential as biomarkers for drug resistance. However, this reviewer has the following concerns.  

Major comments: 

1. In the “3. The role of biomarkers in cancer therapy” section, the author introduced diagnostic biomarkers, as the detection biomarker and the biomarker for the ability to reflect different pathological phenomena (tumor size, invasion degree, and patient survival rate). However, the biomarkers for both expected and adverse therapeutic effects are not mentioned, here. It should be introduced in this section. 

2. From “4. Genes associated with doxorubicin resistance” to “8. Genes associated with nivolumab and pembrolizumab resistance”, the author described that potential resistance marker genes for each drug, but whether these biomarkers work only in liver cancers or also work in other cancers is unknown. 

3. From “4. Genes associated with doxorubicin resistance” to “8. Genes associated with nivolumab and pembrolizumab resistance”, the author described several genes, but the rate of genetic alteration in each gene is not mentioned at all.

4. In “4. Genes associated with doxorubicin resistance”, Two genes, NAP1L1 and CKLF1, and others are described as candidate genes. Although they described that NAP1L1 expression is associated with aggressive clinicopathological features and that CKLF1 might promote malignant transformation, how these genes have to do with doxorubicin resistance is missing. Similarly, in section 5, miR-32-5p is described as overexpressed in 5-FU-resistant HCC, but the reason or mechanism for resistance is not mentioned. In section 6, CD13 in CSCs is correlated with poor prognosis in HCC, but how CD13 promotes HCC progression and induces sorafenib resistance is obscured.

Author Response

Open Review 2

  1. In the “3. The role of biomarkers in cancer therapy” section, the author introduced diagnostic biomarkers, as the detection biomarker and the biomarker for the ability to reflect different pathological phenomena (tumor size, invasion degree, and patient survival rate). However, the biomarkers for both expected and adverse therapeutic effects are not mentioned, here. It should be introduced in this section. 

We have added relevant content.

Page.6, lines.242-246; Page.7, lines.247-249

  1. From “4. Genes associated with doxorubicin resistance” to “8. Genes associated with nivolumab and pembrolizumab resistance”, the author described that potential resistance marker genes for each drug, but whether these biomarkers work only in liver cancers or also work in other cancers is unknown. 

We have added this information between paragraphs 4 to 8.

Page.7-Page.15

  1. From “4. Genes associated with doxorubicin resistance” to “8. Genes associated with nivolumab and pembrolizumab resistance”, the author described several genes, but the rate of genetic alteration in each gene is not mentioned at all.

We have added related content.

Page.7, lines.250-263

  1. In “4. Genes associated with doxorubicin resistance”, Two genes, NAP1L1 and CKLF1, and others are described as candidate genes.

Although they described that NAP1L1 expression is associated with aggressive clinicopathological features and that CKLF1 might promote malignant transformation, how these genes have to do with doxorubicin resistance is missing.

We have added related content.

Page.7, lines.294-298; Page.8, lines.311-312

Similarly, in section 5, miR-32-5p is described as overexpressed in 5-FU-resistant HCC, but the reason or mechanism for resistance is not mentioned.

We have added related content.

Page.9, lines.385-391

In section 6, CD13 in CSCs is correlated with poor prognosis in HCC, but how CD13 promotes HCC progression and induces sorafenib resistance is obscured.

We have added related content.

Page.10, lines.435-444

The manuscript has been revised by a native English speaker.

Round 2

Reviewer 1 Report

The authors have responded to my previous comments.

Reviewer 2 Report

All the comments have been addressed.